# Reflector-Guided Localisation of Non-Palpable Breast Lesions: A Prospective Evaluation of the SAVI SCOUT^®^ System

**DOI:** 10.3390/cancers13102409

**Published:** 2021-05-17

**Authors:** Umar Wazir, Iham Kasem, Michael J. Michell, Tamara Suaris, David Evans, Anmol Malhotra, Kefah Mokbel

**Affiliations:** 1The London Breast Institute, Princess Grace Hospital, London W1U 5NY, UK; umarkhanwazir@gmail.com (U.W.); imm32@cam.ac.uk (I.K.); michael.michell@nhs.net (M.J.M.); t.suaris@nhs.net (T.S.); David.Evans@bartshealth.nhs.uk (D.E.); anmolmalhotra@nhs.net (A.M.); 2Department of General Surgery, Khyber Teaching Hospital, Peshawar 25000, Pakistan

**Keywords:** breast cancer, localisation, occult, SAVI SCOUT^®^, non-palpable breast lesions, reflector-guided localisation

## Abstract

**Simple Summary:**

Marking impalpable areas of breast cancer prior to surgery is an important part of the modern treatment of breast cancer. Traditionally, the target lesion would be marked by a wire just before surgery under image guidance and would help the surgeon locate the tumour during surgery. However, this method has some drawbacks, such as patient discomfort, the risk of migration and dislodgement, and the need to couple surgical and radiological schedules. Therefore, there has been a growing interest in this system, thus supporting its potential. In this study, we have evaluated one such system, SAVI SCOUT^®^, in 63 consecutive patients. Our experience with this system supported its potential role in modern breast surgery.

**Abstract:**

Wire-guided localisation (WGL) has been the mainstay for localising non-palpable breast lesions before excision. Due to its limitations, various wireless alternatives have been developed. In this prospective study, we evaluate the role of radiation-free wireless localisation using the SAVI SCOUT^®^ system at the London Breast Institute. A total of 72 reflectors were deployed in 67 consecutive patients undergoing breast conserving surgery for non-palpable breast lesions. The mean interval between deployment and surgery for the therapeutic cases was 18.8 days (range: 0–210). The median deployment duration was 5 min (range: 1–15 min). The mean distance from the lesion was 1.1 mm (median distance: 0; range: 0–20 mm). The rate of surgical localisation and retrieval of the reflector was 98.6% and 100%, respectively. The median operating time was 28 min (range: 15–55 min) for the therapeutic excision of malignancy and 17 min (range: 15–24) for diagnostic excision. The incidence of reflector migration was 0%. Radial margin positivity in malignant cases was 7%. The median weight for malignant lesions was 19.6 g (range: 3.5–70 g). Radiologists and surgeons rated the system higher than WGL (93.7% and 98.6%, respectively; 60/64 and 70/71). The patient mean satisfaction score was 9.7/10 (*n* = 47, median = 10; range: 7–10). One instance of signal failure was reported. In patients who had breast MRI after the deployment of the reflector, the MRI void signal was <5 mm (*n* = 6). There was no specific technique-related surgical complication. Our study demonstrates that wire-free localisation using SAVI SCOUT^®^ is an effective and time-efficient alternative to WGL with excellent physician and patient acceptance.

## 1. Introduction

Non-palpable breast lesions form a large portion of treated breast lesions. Breast conserving surgery (BCS) has been the dominant paradigm in the surgical treatment of breast neoplastic lesions for the last four decades. Especially in non-palpable lesions, some form of pre-operative localisation of the lesion is often crucial to BCS [1]. The earliest modality for localising impalpable lesion was wire-guided localisation (WGL), which has remained the mainstay for localising non-palpable breast lesions before excision [2,3].

However, WGL does have its drawbacks. It is uncomfortable for the patient and carries a minor risk of needle-stick injury to the surgeon. The wire cannot be retained for prolonged periods. Therefore, slots for radiological insertion of the wire need to be coordinated within 24 h of the resectional surgery. This poses a significant strain on ongoing services, as well as an impediment to the establishment of new services. Wires could migrate or fracture, making it difficult to identify the lesion marked for excision [4,5,6]. Diathermy burns may occur during surgery as heat or current could be conducted to the skin surface [7]. Due to these and other limitations, various alternatives have been developed in search of more optimal methods of non-palpable breast lesion localisation without the limitations of WGL.

The earliest attempts involved the use of radioactive ^125^I seeds implanted under radiological guidance [8,9]. However, the use radioactive seed localisation (RSL) comes with a significant regulatory burden inherent to the use of radioactive materials, which doubtlessly has dissuaded its wider adoption [10,11]. This has encouraged the development of novel non-radioactive, wire-free localisation methods. Examples of devices currently available include LOCalizer™ (Hologic Inc., Santa Carla CA, USA), which makes use of radiofrequency identification tags (RFID) [12]; Magseed^®^ (Endomag limited, Cambridge, UK), which uses a ferro-magnetic tracer to direct a hand-held magnetometer [13], and SAVI SCOUT^®^ (Merit Medical, Aliso Viejo, CA, USA), which utilises a proprietary electromagnetic wave reflector (Figure 1) [14].

In the SAVI SCOUT^®^ localisation system, the reflector can be inserted within or near the targeted breast tissue lesion using a bespoke delivery system. This can be conducted under ultrasound or mammogram guidance in the radiology suite. The reflector can later be localised during surgery by an infra-red emitting detector system which provides a distance to target reading (Figure 2). Successful extraction of the reflector can be confirmed during surgery using the detector or specimen radiographs. The reflector can also be used to mark axillary lymph nodes for targeted dissection and has been approved by the FDA for implantation for an unlimited pre-operative period [15].

In this prospective study, we evaluate the role of radiation-free wireless localisation using the SAVI SCOUT^®^ system at the London Breast Institute (the Princess Grace Hospital, London, UK).

## 2. Materials and Methods

### 2.1. Setting

This technique was evaluated in a prospective cohort of 67 patients undergoing therapeutic excision of non-palpable breast cancer (*n* = 57; median age = 53; range: 27–81 years), or diagnostic excision of screen-detected indeterminate/benign breast abnormality (*n* = 10: median age = 45.5; range: 28–69 years). Of the 57 patients undergoing therapeutic excision, 44 presented with screen-detected malignancy and 13 had non-palpable tumours after neoadjuvant chemotherapy. All patients had triple assessment, including imaging-guided core biopsy, prior to surgery.

This study was performed at the London Breast Institute within the Princess Grace Hospital (London, UK). Patients who required the excision of non-palpable breast lesions were recruited and decisions regarding treatment were made following discussion within the multidisciplinary team.

The radiologist and surgeons of this centre involved in this study have had years of experience with the standard WGL techniques. Furthermore, both LOCalizer™ and Magseed^®^ were evaluated recently for use in our practice [16].

This is a prospective observational clinical evaluation. Therefore, formal ethical approval was deemed unnecessary. However, the use of these technologies was approved by the institution’s multidisciplinary breast cancer board as well as the clinical governance team of the Princess Grace Hospital. All participants signed an informed written consent and detailed patient information leaflets regarding this novel wireless technique were provided.

The primary end points were: 1. rate of successful localisation and retrieval; 2. frequency of positive radial surgical margins for malignant lesions requiring surgical re-excision. Surgical margins were considered positive if the tumour was less than 1 mm away from the nearest radial margin for invasive cancer and 2 mm for pure ductal carcinoma in situ (DCIS); 3. complications specifically related to the SAVI SCOUT^®^ system.

All patients had a control normal mammography film following deployment (Figure 3) and specimen mammography (Figure 4) following surgical excision. These images were used to evaluate the reflector distance from the lesion and migration.

### 2.2. Data Collection

Data were also collected regarding: 1. duration of radiological deployment of reflector; 2. distance between reflector and target lesion; 3. duration of surgical procedure; 4. incidence of migration of reflector; 5. patient and physician acceptance; 6. weight of surgical specimen.

## 3. Results

SAVI SCOUT^®^ reflectors (*n* = 72) (Figure 4) were deployed under ultrasound (*n* = 66) or stereotactic X-ray guidance (*n* = 5) in 67 consecutive patients including 57 patients presenting with non-palpable breast malignancy at the time of diagnosis (*n* = 44) or after neoadjuvant chemotherapy (*n* = 13) up to 210 days prior to surgery (Table 1 and Table 2).

The mean interval between deployment and surgery for the therapeutic cases was 18.8 days (range: 0–210).

A total of 72 reflectors were deployed in 67 patients (five patients had two reflectors).

The nearest distance between the reflectors was 17 mm.

The median deployment duration was 5 min (range: 1–15 min), with a mean distance from the breast lesion of 1.1 mm and a median distance of 0 (range: 0–20 mm).

The rate of successful surgical localisation and retrieval of the reflector was 98.6% and 100%, respectively.

The median operating time including identification and retrieval was 28 min (range: 15–55 min) for therapeutic excision and 17 min (range: 15–24) for diagnostic excision.

A total of six patients had breast MRI after the deployment of the reflector and the MRI void signal was smaller than 5 mm in all cases (Figure 5).

The pathological tumour size ranged from 0 (ypT0) to 67 mm (T3). In patients undergoing therapeutic excision for malignancy (*n* = 57), four (7%) required reoperation for positive radial surgical margins. In all four cases, preoperative imaging underestimated the tumour size by more than 100% and three out of four patients had no residual disease in the re-excision specimens. The median weight for malignant lesions was 19.6 g (range: 3.5–70 g).

### 3.1. Radiologists’ and Surgeon’s Acceptance

Radiologists’ responses were available for 64 procedures. Radiologists rated the SAVI SCOUT^®^ system as much better/better than WGL in 93.7% (60/64) of cases. The surgeon involved rated the technique as better/much better in 98.6% (70/71) of cases.

The SAVI SCOUT^®^ system was also preferred to Magseed^®^ and LOCalizer™ by radiologists (*n* = 6) and the breast surgeon. The mean radiologists rating was 8 (range: 7–9) for SAVI SCOUT^®^, 7 (range: 6–8) for Magseed^®^ and 5.2 (range: 4–6) for LOCalizer™. The surgeon’s rating was 8, 7 and 6, respectively, with 5 being equal to wire-guided localisation.

### 3.2. Patients’ Acceptance

Patients’ feedback was obtained from 43 (64%) patients and the mean satisfaction score was 9.7 out of 10 (median = 10; range: 7–10).

Post-operative pain scores were obtained from a pilot group of 13 patients (mean = 2.8/10; range: 0–6).

### 3.3. Retrieval and Migration

The migration rate, defined as movement of SAVI SCOUT^®^ reflector after deployment by more than 5 mm, was 0%.

Although all SAVI SCOUT^®^ reflectors were retrieved and the target lesion was removed (100%), the signal could not be detected in the operating room in one case. It was superficially placed and was located by palpation. The deactivation of the superficially located SAVI SCOUT^®^ reflector by the surgical lights system was considered as a possible cause of this failure (Figure 3).

### 3.4. Complications

There were no specific SAVI SCOUT^®^-related surgical complications.

There was one patient who developed a hematoma that required evacuation and another patient developed a minor wound dehiscence that was sutured under local anaesthesia in the outpatient setting.

## 4. Discussion

As stated earlier, the localisation of occult lesions within the breast is crucial for much of BCS. This is predominantly achieved by WGL, which has been in mode since its initial description in the 1970s and remains the current standard of care [2,17].

The significant limitations of WGL highlighted above have stimulated interest in developing alternatives that overcome the aforementioned limitations.

An early alternative was radio-labelled seed localisation (RSL). In this technique, a titanium seed containing ^125^I is implanted at the site of the lesion and is located intra-operatively with a gamma camera. This had shown considerable promise with regard to patient acceptability, learning curve, and accuracy in the form of low margin positivity [18]. Furthermore, as the seed could be retained safely for up to 5 days, the scheduling requirements were less onerous with RSL compared to WGL [10]. However, the expense and regulatory burden associated with the use of radioactive agents are important practical considerations [19]. Whilst exceedingly unlikely, fracturing of retained radio-labelled seeds during histopathological slicing has been reported in the literature, and would have to be accounted for in workplace guidelines and procedures [6]. Together, these limitations probably dissuade the wide-spread adoption of RSL.

In recent years, there have been several developments of wireless non-radioactive localisation techniques for breast lesions. An obvious advantage of such modalities would be the possibility of leaving the marker in situ for longer periods of time, easing pressures on the scheduling of the procedures. However, the devices available are proprietary and are currently being evaluated in different jurisdictions [20]. Some use ferro-magnetic seeds or solutions as tracers coupled with a handheld magnetometer, such as Magseed^®^ and MaMaLoc (Sirius Medical, Eindhoven, the Netherlands) [13,21]. Another wire-free system, LOCalizer^TM^, uses radiofrequency identification tags [22].

SAVI SCOUT^®^ uses an electromagnetic wave reflector which reflects a combination of infra-red and radar signals from a handheld detector. It has been approved by the FDA for long-term implantation [15].

Our study provides further evidence supporting the reliability of the SAVI SCOUT^®^ system in guiding breast conserving surgery and confirms the benefits of the technology in advancing patient care. Our findings by and large coincide with the findings of Srour et al., with no observed device-specific complications, migration of device, and favourable deployment and retrieval rates and durations [23]. We observed a margin positivity rate of 7%, which is significantly lower than the 22% re-excision rate observed in the UK NHS Breast Screening Programme [24]. However, our series included, in addition to screen-detected cancers, 11 patients who received neoadjuvant systemic therapy (NST) to downstage the disease.

Preoperative imaging underestimated the tumour size by more than 100% in all four cases with positive margin. This suggests that the margin positivity rate we observed may be unrelated to the localisation technique used. In addition, our recently conducted pooled analysis reported a pooled analysis overall re-excision rate of 12.9% for SAVI SCOUT^®^ versus 21% for WGL (relative risk = 0.61) in studies comparing the two methods [25].

When taken in context with the relatively low specimen weight (19.5 g), the low rate of margin positivity is evidence of the accuracy of SAVI SCOUT^®^ as a localisation modality compared to WGL. This aspect, as well as issues pertaining to learning curves and inter-operator variability, would be a worthy subject of future investigations.

Within our current study cohort, the median duration for reflector deployment was found to be 5 min, half the time reported for wire placement during WGL (10 min) [17]. Our study also found that reflectors were placed at the target tissue site at a mean interval of 15.3 days before surgery. Srour et al. compared SAVI SCOUT^®^ with WGL across a range of time-specific variables [23]. Their results showed that WGL is associated with significantly longer delays in surgical operation starting times, as well as longer preoperative times on the day of surgery. These findings vindicate the *raison d’etre* for the novel wire-free localisation systems, such as SAVI SCOUT^®^, allowing for the decoupling of radiology and surgical scheduling, potentially resulting in more efficient use of healthcare facilities, labour and capital.

Our study enables us to compare our experience with this technology to the other radiation-free wireless localisation systems for non-palpable breast lesions. A recent systematic review and pooled analysis conducted by our institute [26] found the Magseed^®^ localisation system to have similarly high successful placement (94.42%) and retrieval rates (99.86%) as the SAVI SCOUT^®^ system (99.64% and 99.64%, respectively) [25]. However, the margin positivity rate observed in our study is significantly lower than that reported by a similar institution using Magseed^®^ (24%), despite a similar median specimen weight of 19.5 g [27].

Several other more qualitative advantages of the SAVI SCOUT^®^ system over Magseed^®^ localisation are also evident. The SAVI SCOUT^®^ detector console gives the distance from the reflector, which is a glaring omission in the case of the Magseed^®^ system. Magseeds^®^ can only be detected up to a depth of 4 cm from the skin surface whilst the SAVI SCOUT^®^ reflector is recommended for use up to 6 cm depths [28]. This allows for the localisation of deeper lesions, resulting in greater usability of the technology. Furthermore, we have found that the device did not interfere with MRI scans in any significant way. This is in line with previous studies, which have reported minimal MRI signal interferences by the SAVI SCOUT^®^ reflector, measuring <5 mm [29]. In contrast, RFID tags and Magseeds^®^ have been associated with significant MRI signal void artefacts, which may impede the monitoring and identification of residual disease on follow-up MRI scans [29]. Furthermore, the SAVI SCOUT^®^ reflector could be deployed at the time of biopsy in patients undergoing NST as it would not interfere in the monitoring of disease response to treatment by imaging. This could save patients a second procedure and enable the more efficient utilisation of hospital resources. The SAVI SCOUT^®^ reflector is also more easily visualised on ultrasonography than magnetic seeds or RFID tags.

When compared to LOCalizer™, SAVI SCOUT^®^ has a few salient advantages. Compared to SAVI SCOUT^®^, LOCalizer™ has a much larger MRI void artefact compared to SAVI SCOUT^®^ (20 mm vs. 5 mm), which limits LOCalizer™’s utility in the neoadjuvant setting. SAVI SCOUT^®^ has a smaller introducer compared to LOCalizer™ (16 gauge compared to 12), which may have an effect on patient acceptance and deployment accuracy [20].

Importantly, SAVI SCOUT^®^ was associated with a high degree of patient satisfaction. This is likely due to several factors, including the ability to deploy the reflector at diagnostic biopsy, the lack of time delays on the day of surgery and the minimal discomfort associated with deployment.

Radiologists and surgeons preferred SAVI SCOUT^®^ to WGL, as well as other novel localisation technologies which were recently in use at our institution (LOCalizer^TM^ and Magseed^®^). It was preferred to Magseed^®^ in view of the need to remove all metal surgical tools from the surgical field when the attempting to localise the Magseed^®^ using the detection probe [5]. The other influencing factor was the significant MRI void signals associated with Magseed^®^ biopsy in patients considered for NST [26].

The SAVI SCOUT^®^ localisation system has also been approved for use in targeted axillary lymph node dissection (TAD) (Figure 6). Although published data are limited, several existing studies have reported the successful use of the reflector system to localise axillary lymph nodes [30,31,32]. Sun et al. reported the successful retrieval of all SAVI SCOUT^®^ reflectors with no intra- or postoperative complications [32]. The successful deployment of SAVI SCOUT^®^ to mark and excise lymph nodes with minimal MRI signal interference suggest that the SAVI SCOUT^®^ system is an effective technology for use in TAD following NST.

We observed only one instance of signal loss. Interestingly, it was in a device inserted closer to the skin than usual (Figure 3). This is in contrast to the literature, which reports that the signal is more likely to fail at depths greater than 4 cm [33]. We believe this signal failure may have been due to overhead theatre lights interfering with the device and we therefore initially recommend that the reflector is deployed behind the lesion for superficially located masses. However, further research into this is required.

Several other limitations of the SAVI SCOUT^®^ system warrant mention. Although rare, the reflector cannot be used in patients who are allergic to nickel, somewhat limiting its use. Furthermore, initial research reported that the deactivation of the reflector may occur when it encounters surgical electrocautery. However, the manufacturer has since made modifications to overcome this limitation [15]. In addition, antenna transection has been reported during surgical dissection, again potentially resulting in signal loss [34]. In either scenario, the signal loss is unlikely to pose an issue to the operating surgeon as the reflector would be in close proximity and could be palpated easily. Calcified fibroadenomas and post-biopsy haematoma have been described as causing issues with signal detection within the operating theatre [31,35]. Previous research has suggested that this may be overcome by placing the reflector next to the haematoma, rather than within the mass [31].

As a prospective cohort study, we believe our data provide good evidence for strongly considering SAVI SCOUT^®^ as an alternative to WGL in the management of occult breast lesions. Ours is the first study evaluating this system in Europe. We believe this technology should be further evaluated in comparison to other novel localisation systems as well as WGL and RSL, in order to better establish its non-inferiority to current gold standards.

The limitations of our study include a relatively small sample size (67 reflectors). However, our findings contribute to the growing body of the literature and are consistent with our recently published pooled analysis of 842 reflectors [25]. Furthermore, this study was conducted in an academic facility, and involved a senior academic breast surgeon and several dedicated breast radiologists. These findings may not be universally generalisable and should be replicated in more representative tertiary care centres.

Moreover, we did not perform a direct head-to-head comparison with WGL, Magseeds^®^ or LOCalizer^TM^. Prospective studies including these modalities would be essential to inform best practice.

The benefits in terms of the decoupling of surgical and radiological timeslots and reduced patient discomfort should more than offset the costs in the medium term. This is indeed an issue which has not thus far been adequately addressed [26]. Good quality evidence pertaining to this issue would have an appreciable effect on the quality of care we could offer to breast cancer patients in the future.

## 5. Conclusions

Our study demonstrates that wire-free localisation using SAVI SCOUT^®^ is a viable and promising alternative to WGL with excellent physician and patient acceptance.

## Figures and Tables

**Figure 1 cancers-13-02409-f001:**
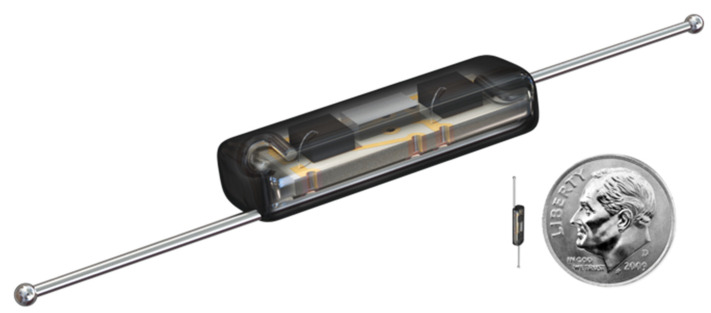
The SAVI SCCOUT^®^ reflector (12 mm long, deployable by a 16 GA needle), with a coin for comparison.

**Figure 2 cancers-13-02409-f002:**
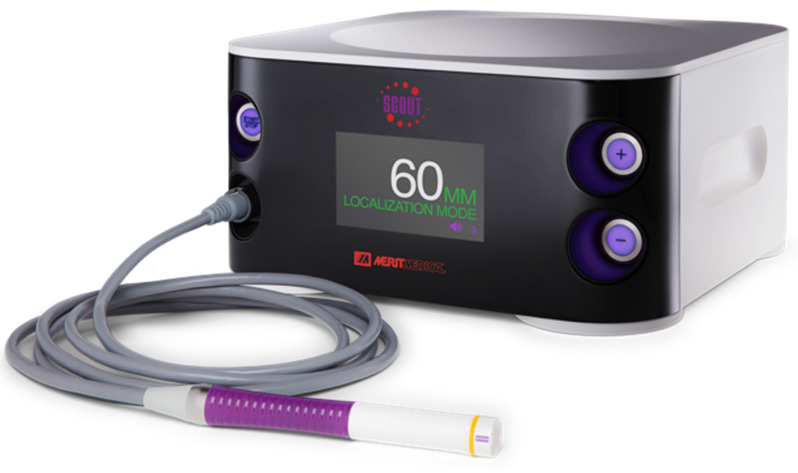
The SAVI SCOUT^®^ handheld detector system with handpiece and integrated console display.

**Figure 3 cancers-13-02409-f003:**
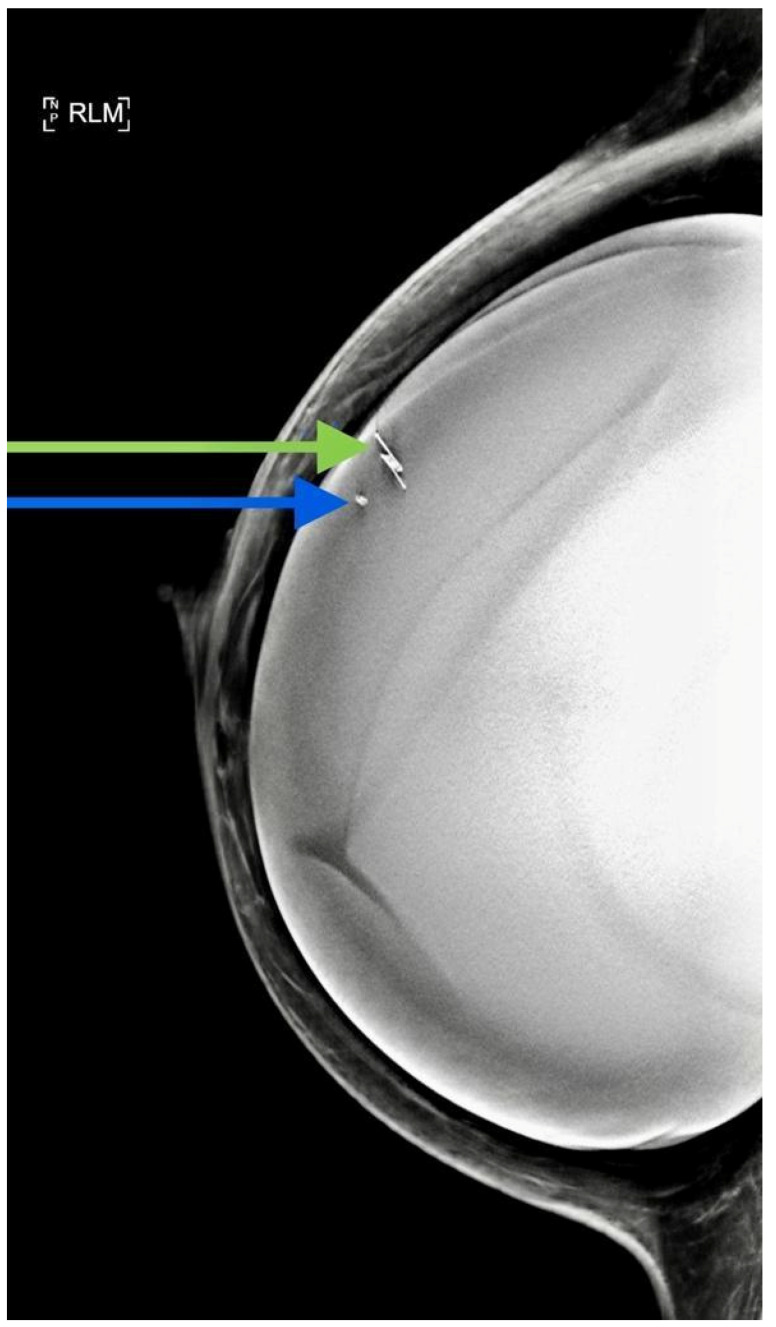
A control mammogram after deployment of the SAVI SCOUT^®^ reflector (green arrow) in a patient with a silicon implant. Blue arrow points to a marker clip.

**Figure 4 cancers-13-02409-f004:**
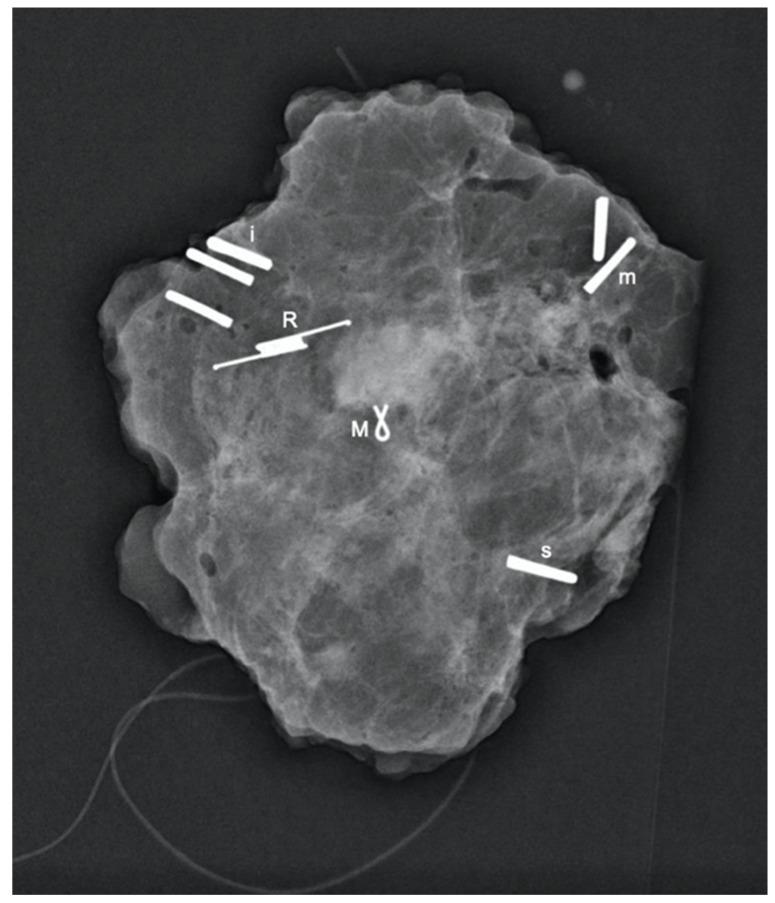
Breast resection specimen with SAVI SCOUT^®^ reflector (R) and marker clip (M) adjacent to the tumour. Peripheral metallic clips can also be seen (i, m and s) for orientation.

**Figure 5 cancers-13-02409-f005:**
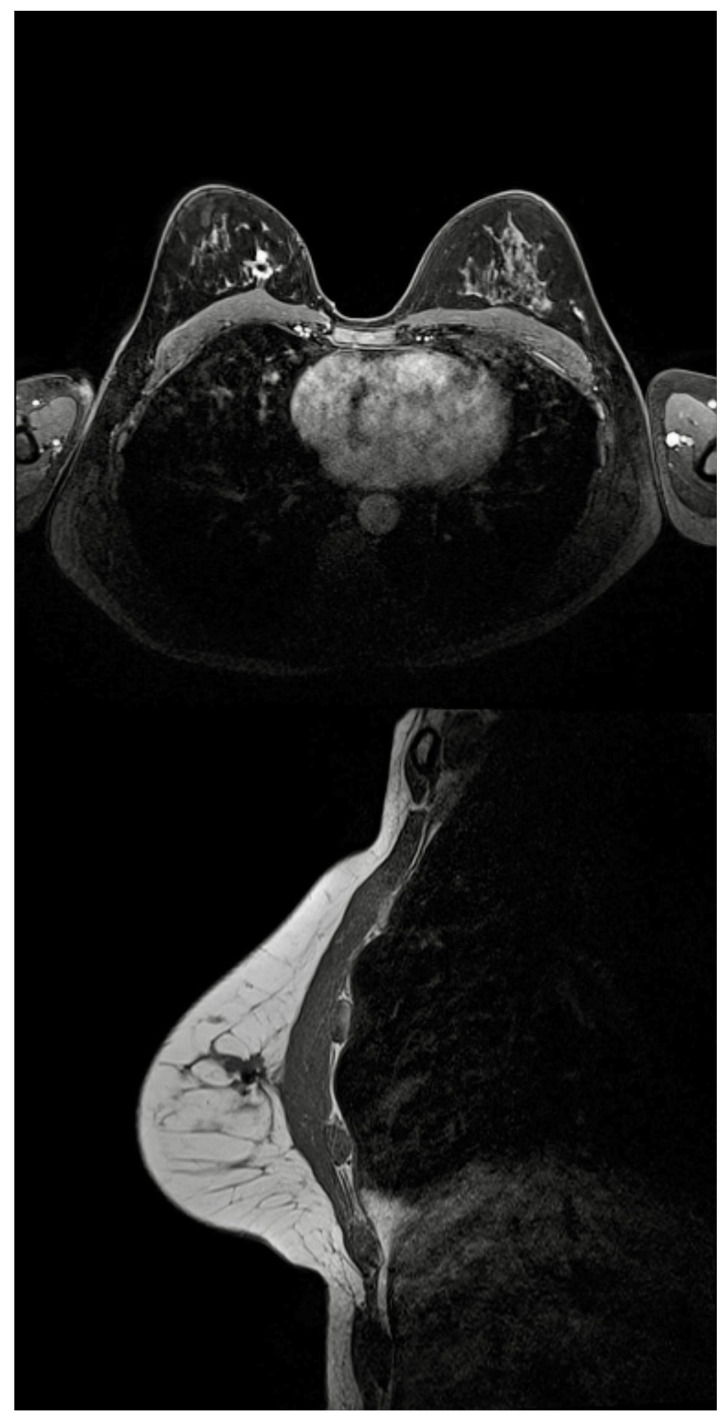
Breast MRI demonstrating a good partial response to NST for TNBC. It demonstrates a small (4.6 mm) MRI void signal generated by the SAVI SCOUT^®^ reflector located within the residual tumour (in the upper inner quadrant of the right breast) that decreased.

**Figure 6 cancers-13-02409-f006:**
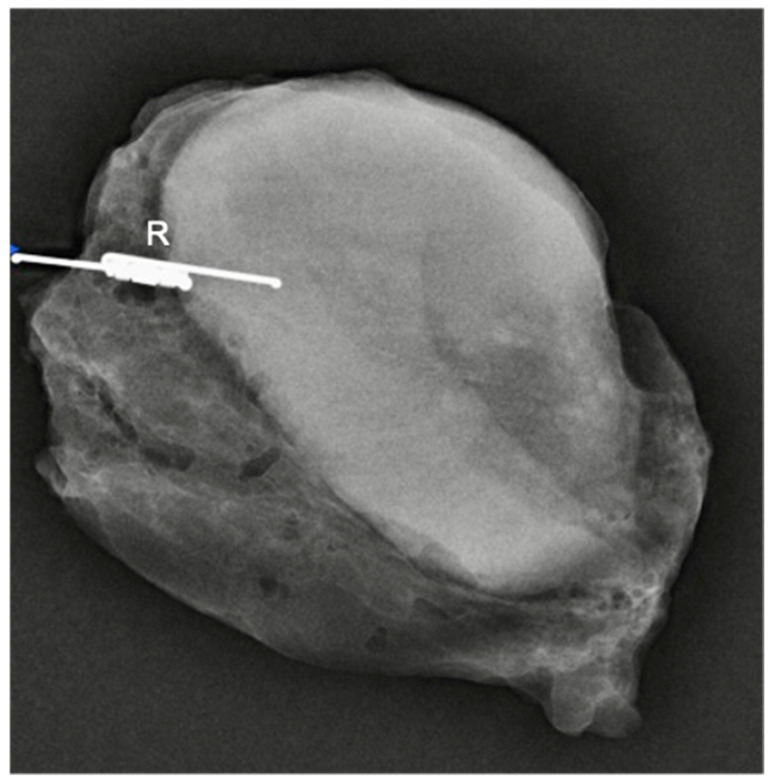
A specimen radiograph demonstrating a SAVI SCOUT^®^ reflector (R) marking a lymph-node, harvested after neoadjuvant systemic therapy (NST) as part of targeted axillary dissection.

**Table 1 cancers-13-02409-t001:** Pathological characteristics and localisation details in therapeutic excision of non-palpable cancer cases. (IDC: invasive ductal carcinoma; DCIS: ductal carcinoma in situ; TNM: tumour/node/metastasis staging).

Case	Age	Distance of Scout to Target on Mammogram (mm)	Duration of Localisation (Minutes)	Duration of Identification and Retrieval (Minutes)	Specimen Weight (g)	Radiologist’s Feedback Compared with Wire Localisation	Patient’s Satisfaction	Pathological TNM Stage	Radial Margins’ Status	Time Interval between Localisation and Surgery (Days)
1	52	0	5	31	20	neutral	10	pT1c (multi) N0	clear	4
2	50	0	5	29	31	better	10	pTis 41 mm	clear	3
3	46	0	5	15	14.5	much better	9	pT1bN0	clear	1
4	55	0	5	50	25	better	10	pT1cN0	clear	1
5	58	0	5	23	19	much better	9	ypT2 (multi) N1mic	clear	1
		0	5						clear	
6	66	0	4	15	12	much better	10	pT1bN0	clear	1
7	74	3	5	28	10.5	better	10	pT2 N0	clear	1
8	60	0	3	25	21.5	much better	10	pT2N1	clear	0
9	74	0 (stereotactic)	10	38	39.5	better	9	pT1aN0	clear	7
10	53	0	15	40	34	better	10	pT1cN0	clear	1
11	76	0	2	19	7.5	-	10	pTis	clear	
12	50	0	3	45	11.5	better	10	ypT1N0	clear	34
13	59	0	2	23	70	better	10	T1cN0	clear	0
14	27	3	3	26	16	better	10	ypT1aN0	clear	0
15	45	3	5	20	14.5	much better		ypT0N0	clear	8
16	36	0	4	37	19.5	better	7	T3N0	focally positive for DCIS	0
17	62	4.5	15	22	13	better	10	Tis	clear	0
18	53	2	8	30	13	better	9	ypT0N0	clear	7
19	63	0	4	25	32	neutral	10	pT1N0	clear	0
20	42	0	3	28	31.5	better	10	pT2NxM1	clear	0
21	60	0	3	30	9.5	better		pTis	clear	14
22	53	2	8	30	13	better	10	ypT0N0	clear	19
23	74	3	5	30	17	neutral	10	pTisN0	clear	0
24	48	1	5	30	32	much better	9	pT1N0	clear	1
25	59	0	5	30	51.5	better	10	pT3N1	positive	0
		0	5	30	38.5	better		pT1N0		
26	36	0	5	28	-	better	10	pT1cN0	clear	0
27	53	0	5	25	23	much better	10	pT1cN0	clear	11
28	40			30	4.5	-		pT2N0	positive	5
29	47	0	7	29	14	neutral	10	pT1cN0	clear	3
30	32	0	5	25	13	better	9	ypT0N0	clear	210
31	47	0	3	23	7.5	better	10	ypTisN0	clear	101
32	63	20 (stereotactic)	10	25	-	better	10	pTis	clear	3
33	44	0	10	26	-	much better	10	pT1N0 multifocal	clear	3
34	58	0	5	38	32.5	much better	8	pT2N0	clear	2
35	42	0	5	22	-	-		pTis (16 mm)	clear	0
36	46	16 (stereotactic)	5	55	8	less favourable than wire	10	pT1	clear	0
37	46	0	5	38	26.5	better	10	pT1N0	clear	0
38	64	0	2	35	20	better	10	pT1N0	clear	3
39	58	0	3	22	-	better		pT2N0	clear	5
40	51	0	6	32	35	-		pTis (21 mm)	positive (1 mm: radial margin)	2
41	52	0	5	31	33.5	better	10	ypTisN0	clear	1
42	73	0	2	25	34.5	-	10	ypT0N0	clear	97
43	55	0	5	35	68.5	better		pT2N0	clear	5
44	55	0	10	21	38	much better	10	pT1N0 multifocal	clear	24
		15	10	21						
45	55	0	10	17	14.4	much better		pT1b (multi) N1 (micro)	clear	3
46	64	0	3	25	5.3	much better	10	ypT0N0	clear	197
47	45	0	-	19	19.8	better		pT1N0	clear	8
48	78	0	3	16	10.5	-		pT1N0	clear	19
49	41	0	2	20	7.5	better	10	pTis	clear	3
50	71	0	3	19	3.5	much better	10	pT1cN0	clear	8
51	62	0	5	34	27	much better	10	pTis (multi)	clear	36
		Stereotactic								
52	80	0	2	26	56	much better	10	pT1c (multi) N0	clear	3
		0	2							
53	81	0	3	15	14.5	better		pT1bN0	clear	3
54	71	-	5	17	17.5	better		ypTisN2M1	clear	3
55	48	0	5	30	-	-		pT2N0	clear	3
56	50	0	3	45	24	much better		pT2N0	clear	1
57	50	0	5	35	24	-		ypT1N0	clear	188

**Table 2 cancers-13-02409-t002:** Pathological characteristics and localisation details in diagnostic cases.

Case	Age	Distance of Scout to Target on Mammogram (mm)	Duration of Localisation (minutes)	Duration of Identification and Retrieval (minutes)	Specimen Weight (g)	Radiologist’s Feedback Compared with Wire Localisation	Patient’s Satisfaction	Pathology	Time Interval between Localisation and Surgery (Days)
1	58	0	1	18	7.5	much better	10	Papilloma	0
2	47	0	4	19	11.5	much better	10	Benign breast change	0
		0	4	19	13.5			Papilloma	
3	69	0	15	15	18.5	better		Malignant adenomyoepithelioma + papillomatosis	0
4	68	0	3	16	-	much better		ADH	0
5	48	0	5	15	7.5	neutral		CSL	
6	28	0	5	15	3	better		Fibroadenoma	0
7	34	1 (stereotactic)	5	24	11	-		LCIS/ALH/CSL	0
8	36	1	5	17	5	better		papilloma	0
9	38	0	4	16	8	better		Atypical columnar change	3
10	44	0	5	17	18	better	10	CSL	0

## Data Availability

The data presented in this study are available in this article.

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
