# Peer review of "Reflector-Guided Localisation of Non-Palpable Breast Lesions: A Prospective Evaluation of the SAVI SCOUT® System"

_cancers, 2021, doi:10.3390/cancers13102409_

Round 1
Reviewer 1 Report
interesting method, well presented
Author Response
- “…interesting method, well presented.”
Thank you for your support of our manuscript. We are very pleased that you find our work worthwhile and value your encouragement.
Umar Wazir
Reviewer 2 Report
General Comments
This paper is well written and the authors' logic is easy to follow. My primary concern with the article is that the SAVI SCOUT system is not compared with any of the other localisation systems directly. The authors do provide some subjective comparisons on page 10, paragraph 2, and they also acknowledge this limitation on page 13, paragraph 2. That said, their conclusion about the benefits of SAVI SCOUT would be far stronger if there had been a head-to-head comparison.
They do state that the LOCalizer system from Hologic may have a cost advantage over the SAVI SCOUT system (see page 12, paragraph 2 and page 13 paragraph 3) but there is no mention of actual numbers. If I was a surgeon contemplating the use of SAVI SCOUT, how much more would the patient or healthcare system have to pay?
When an acronym is used for the first time, it needs to be spelled out in full. For example, what does SSL on page 5 (last line) stand for? Is it Secure Socket Layer? I think not! Similarly NST is introduced for the first time on page 11 (paragraph 3) but is only defined at the top of page 12.
Specific Comments
page 1 line 13 However, this method has some drawbacks, such as ...
page 1 line 16 ... this system thus supports its potential role ...
page 3 line 8 ... the use of this technology was approved by ...
page 3 line 15 ... is the tumour was less than 1 mm away ...
page 5 line 7 ... or stereotactic X-ray guidance (n=5) in 63 ...
page 9 line 1 I believe this should be Table 2.
page 11 line 8 ... in advancing patient care.
page 11 line 12 ... Breast Screening Programme [23].
page 12 line 8 ... (20 mm vs. 5 mm)
page 12 line 47 ... next to the haematoma , rather than ...
page 15 line 4 ... academic and clinical advice to ...
Author Response
This paper is well written and the authors' logic is easy to follow.
- My primary concern with the article is that the SAVI SCOUT system is not compared with any of the other localisation systems directly. The authors do provide some subjective comparisons on page 10, paragraph 2, and they also acknowledge this limitation on page 13, paragraph 2. That said, their conclusion about the benefits of SAVI SCOUT would be far stronger if there had been a head-to-head comparison.
We have removed this statement from the conclusion.
- They do state that the LOCalizer system from Hologic may have a cost advantage over the SAVI SCOUT system (see page 12, paragraph 2 and page 13 paragraph 3) but there is no mention of actual numbers. If I was a surgeon contemplating the use of SAVI SCOUT, how much more would the patient or healthcare system have to pay?
We have removed these statements, as a citable price is currently not available.
- When an acronym is used for the first time, it needs to be spelled out in full. For example, what does SSL on page 5 (last line) stand for? Is it Secure Socket Layer? I think not!
We have amended as advised.
- Similarly, NST is introduced for the first time on page 11 (paragraph 3) but is only defined at the top of page 12.
We have amended as advised.
Specific Comments
- page 1 line 13 However, this method has some drawbacks, such as ...
- page 1 line 16 ... this system thus supports its potential role ...
- page 3 line 8 ... the use of this technology was approved by ...
- page 3 line 15 ... is the tumour was less than 1 mm away ...
- page 5 line 7 ... or stereotactic X-ray guidance (n=5) in 63 ...
- page 9 line 1 I believe this should be Table 2.
- page 11 line 8 ... in advancing patient care.
- page 11 line 12 ... Breast Screening Programme [23].
- page 12 line 8 ... (20 mm vs. 5 mm)
- page 12 line 47 ... next to the haematoma, rather than ...
- page 15 line 4 ... academic and clinical advice to ...
We have applied the above recommended amendments (5-15).
Reviewer 3 Report
Title: Reflector-guided Localization of Non-palpable Breast Lesions: A Prospective Evaluation of the SAVI SCOUT® System
Summary: In this prospective study, the Authors evaluated the role of radiation-free wireless localisation us-20 ing the SAVI SCOUT® system; 67 reflectors were deployed in 63 consecutive patients undergoing breast conserving surgery for non-palpable breast lesions. The median deployment duration was 5 minutes (range: 1-15 minutes), with a mean distance from the lesion of 1.3 mm and a median distance of 0 (range: 0-20 mm). Reflector retrieval was 100%. Median operating time was 27 min (range: 15-55 minutes) for therapeutic excision of malignancy and 17 min (range: 15-24) for diagnostic excision. The incidence of reflector migration was 0%. Radial margin positivity in malignant cases was 7.5%. The median weight for malignant lesions was 19.5 g (range: 3.5-70g). Radiologists and surgeons rated the system higher than WGL (93.5% & 98.5% respectively). Only one instance of signal failure was reported. The Authors concluded that this study demonstrated that wire-30 free localisation using SAVI SCOUT® is an effective and time-efficient alternative to WGL with excellent physician and patient acceptance.
General Comment: This paper was focused on a new localization technique, which proved to show some advantages compared to traditional localizations according to literature; literature has been limited to less than 1000 cases, as reported in a recent metanalisis of the same Authors, and new cases could add some data to the existent experience. Nonetheless, the Materials and methods were not described in detail, the lack of localizing techniques comparison and the small population could affect the impact of this study on general knowledge.
New data were provided only for 46 patients, considering that the Authors already used 17 of 63 patients in this previously published paper “Tayeh S, Muktar S, Heeney J, Michell MJ, Perry N, Suaris T, Evans D, Malhotra A, Mokbel K. Reflector-guided Localization of Non-palpable Breast Lesions: The First Reported European Evaluation of the SAVI SCOUT® System. Anticancer Res. 2020 Jul;40(7):3915-3924. doi: 10.21873/anticanres.14382. PMID: 32620632.”, as provided in detail by the same tables in both the papers. I have also ethical concerns about the use of already published data.
Specific comments:
Title: ok
Abstract: ok
Introduction: ok
Materials and Methods:
- Results concerning MRI were not precisely reported (“No significant void signals were observed in relation to SAVI SCOUT® in patients who had breast magnetic resonance imaging (MRI) scans after reflector deployment”) nor assessed in Material and methods. I can understand the importance of this information, and Authors should assess how many patients had MRI and how large the void signal size was.
- Methods for patients’ and clinicians’ acceptance should be explained in Material and methods; in particular which is the asked question for the VAS evaluation to patients.
- Was the distance from the lesion calculated from mammograpy, surgical specimen mammography, or at pathological evaluation?
Results:
- Magseed and Localizer systems, were not mentioned in Material and methods, hence they could not be used as a comparison as no information were reported about clinicians’ experience on those systems.
- As well, if the Authors want to compare the clinicians’ acceptance ot SAVI scout with hookwire localization (WGL), they should at least report the clinicians’ experience on WGL.
- Which way was the undetected reflector (1/67) retrieved, considering that according to Authors all 67 (100%) were retrieved?
Discussion:
- The discussion should be more focused on the study results and less on a general evaluation of SAVI scout system or a comparison with other new techniques not directly studied by the Authors.
- Economic evaluation is not a topic for this study and it could not be discussed unless studied in detail by this paper.
- The conclusion that “wire-free localization using SAVI SCOUT® is an effective and time-efficient alternative to WGL with excellent physician and patient acceptance” should be validate by a more precise methodology, as previusly suggested.
- The final sentence “We believe that our study helps recommend this system as a feasible alternative to current methodologies” should be delated as no scientific experience on merely 67 cases may lead to this recommendation.
Figures: ok
Tables: ok (except for the first 17 patients, see general comment).
Author Response
- General Comment: This paper was focused on a new localization technique, which proved to show some advantages compared to traditional localizations according to literature; literature has been limited to less than 1000 cases, as reported in a recent metanalisis of the same Authors, and new cases could add some data to the existent experience. Nonetheless, the Materials and methods were not described in detail, the lack of localizing techniques comparison and the small population could affect the impact of this study on general knowledge.
We have acknowledged sample size issue in our discussion, and have endeavoured to rectify the problem by revisiting the data, and have thus revised the result section to reflect this.
- New data were provided only for 46 patients, considering that the Authors already used 17 of 63 patients in this previously published paper “Tayeh S, Muktar S, Heeney J, Michell MJ, Perry N, Suaris T, Evans D, Malhotra A, Mokbel K. Reflector-guided Localization of Non-palpable Breast Lesions: The First Reported European Evaluation of the SAVI SCOUT® System. Anticancer Res. 2020 Jul;40(7):3915-3924. doi: 10.21873/anticanres.14382. PMID: 32620632.”, as provided in detail by the same tables in both the papers. I have also ethical concerns about the use of already published data.
As stated above, we have revised the result section, and now the number of reflectors deployed is 72. We acknowledge that the initial 17 cases were reported in our pilot series (se ref 27). We have had significantly more extensive experience since then, and believed that insights would be of interest to the professional community. We understand the ethical concern raised, but fail to see the value in excluding data from the study, especially in view of the concerns regarding numbers raised in point #1. It is reasonable and Ethical to include the previously published pilot study which represented a snapshot of our initial experience and part of our ongoing prospective evaluation provided we cite that study (as we did) and make it clear that it was previously published.
Specific comments:
Title: ok
Abstract: ok
Introduction: ok
Materials and Methods:
- Results concerning MRI were not precisely reported (“No significant void signals were observed in relation to SAVI SCOUT® in patients who had breast magnetic resonance imaging (MRI) scans after reflector deployment”) nor assessed in Material and methods. I can understand the importance of this information, and Authors should assess how many patients had MRI and how large the void signal size was.
We appreciate this valuable advise and have amended the above statement as follow: “6 patients had breast MRI after deployment of reflector and the MRI void signal was smaller than 5 mm in all cases”.
- Methods for patients’ and clinicians’ acceptance should be explained in Material and methods; in particular which is the asked question for the VAS evaluation to patients.
- Was the distance from the lesion calculated from mammograpy, surgical specimen mammography, or at pathological evaluation?
This is addressed in the following statement in the second last paragraph of the materials and methods section: “All patients had a control normal mammography film following deployment (Figure 3) and specimen mammography (Figure 4) following surgical excision.” To further clarify, we added the following: “These images were used to evaluate reflector distance from lesion and migration.”
Results:
- Magseed and Localizer systems, were not mentioned in Material and methods, hence they could not be used as a comparison as no information were reported about clinicians’ experience on those systems.
- As well, if the Authors want to compare the clinicians’ acceptance ot SAVI scout with hookwire localization (WGL), they should at least report the clinicians’ experience on WGL.
We have added the following statement to address the above two pints: “The radiologist and surgeons of this centre involved in this study have had years of experience with the standard WGL techniques. Furthermore, both LOCalizer™ and Magseed® were evaluated for use recently for use in our practice [16].
- Which way was the undetected reflector (1/67) retrieved, considering that according to Authors all 67 (100%) were retrieved?
To explicate the same, we have added the following stated: “It was superficially placed and was located by palpation.”.
Discussion:
- The discussion should be more focused on the study results and less on a general evaluation of SAVI scout system or a comparison with other new techniques not directly studied by the Authors.
This has addressed in point #7, where we added reference 16 demonstrating our previous experience with the alternative techniques.
- Economic evaluation is not a topic for this study and it could not be discussed unless studied in detail by this paper.
We have removed the pertinent statements from our discussion.
- The conclusion that “wire-free localization using SAVI SCOUT® is an effective and time-efficient alternative to WGL with excellent physician and patient acceptance” should be validate by a more precise methodology, as previusly suggested.
We have changed the statement to “…viable and promising…”
- The final sentence “We believe that our study helps recommend this system as a feasible alternative to current methodologies” should be delated as no scientific experience on merely 67 cases may lead to this recommendation.
Done.
Figures: ok
Tables: ok (except for the first 17 patients, see general comment).